# Recent Development of Nanomaterials for Transdermal Drug Delivery

**DOI:** 10.3390/biomedicines11041124

**Published:** 2023-04-07

**Authors:** Moong Yan Leong, Yeo Lee Kong, Kevin Burgess, Won Fen Wong, Gautam Sethi, Chung Yeng Looi

**Affiliations:** 1School of Biosciences, Faculty of Health and Medical Sciences, Taylor’s University Lakeside Campus, Subang Jaya, Selangor Darul Ehsan 47500, Malaysia; 2Department of Engineering and Applied Science, America Degree Program, Taylor’s University Lakeside Campus, Subang Jaya, Selangor Darul Ehsan 47500, Malaysia; 3Department of Chemistry, Texas A&M University, P.O. Box 30012, College Station, TX 77842, USA; 4Department of Medical Microbiology, Faculty of Medicine, Universiti Malaya, Kuala Lumpur 50603, Malaysia; 5Department of Pharmacology, Yong Loo Lin School of Medicine, National University of Singapore, Singapore 117600, Singapore

**Keywords:** nanomaterials, transdermal drug delivery, transdermal mechanisms

## Abstract

Nano-engineered medical products first appeared in the last decade. The current research in this area focuses on developing safe drugs with minimal adverse effects associated with the pharmacologically active cargo. Transdermal drug delivery, an alternative to oral administration, offers patient convenience, avoids first-pass hepatic metabolism, provides local targeting, and reduces effective drug toxicities. Nanomaterials provide alternatives to conventional transdermal drug delivery including patches, gels, sprays, and lotions, but it is crucial to understand the transport mechanisms involved. This article reviews the recent research trends in transdermal drug delivery and emphasizes the mechanisms and nano-formulations currently in vogue.

## 1. Introduction

Targeted medication comprises selective deployment of pharmaceutically active components at predetermined sites to reduce adverse effects and enhance treatment efficacy [1,2,3]; essentially, these methods increase therapeutic indices. Targeted approaches have been significantly advanced by colloidal carrier forms centered on biocompatible and biodegradable polymerics, including nanoparticles (NPs) [4]. According to one definition [5,6], nanoparticles are small solid colloidal substances of 1–1000 nm diameter, made from polymers, lipids, or metals. Active particles in NP may be dissolved, encapsulated, entrapped, or absorbed [7]. Preferred NPs for nanomedicine are loosely regarded as less than 200 nm [8]. Particle size plays a crucial role on drug distribution, release rates, targeting abilities, and toxicities [9]. NPs can be classified according to their properties, shapes, sizes [10], or into nanospheres (matrix structure for dispersing functional biological materials) and nanocapsules (membrane structure using an oil or aqueous core to contain drugs) [11]. 

NPs modify permeability [12], half-life [13], cytotoxicity [14,15], pharmacokinetics of medications, and diagnostic agents [16]. They can be nanosensors [17,18], drug carriers [19,20], and diagnostic agents [21,22]. Research has emphasized improving absorption through the skin to maintain homeostasis [23,24,25] using micelles [26], liposomes [27], or polymers [28]. NPs can infiltrate the skin via intracellular, intercellular, or transcellular pathways [29,30,31] where penetration depths are determined by particle size [32,33] initiated by hair movement within the follicle; 600 nm particle size may be optimal for some NPs [34]; however, permeation will depend on NP composition. Permeability images for 30 nm cadmium selenide/zinc sulfide (CS/ZnSO_4_) NPs through mouse skin after ultraviolet light (UV)-B radiation-induced epidermal disturbance [35] revealed that these NPs in sebaceous glands and in the skin are aggregated. NPs in this study tended to be localized in the epidermis interstitium [35,36,37].

## 2. Important NP Physical Compositions

### 2.1. Micellular

Micelles are amphiphilic spherical structures with hydrophobic and hydrophilic regions [38] varying from 5 to 100 nm [39]. The hydrophilic region of micelle allows intravenous administration, whereas hydrophobic regions tend to be for cargo storage [40]. These micellar NPs can deliver macromolecules as they offer a sustained and controlled release of biomolecules, physicochemical stability of the embedded molecules, enhanced drug pharmacokinetics, and drug bioavailability [41]. Micellar NPs promote small NPs, good entrapment efficiency, and are inexpensive relative to other nanocarriers (liposomes and niosomes) [42]. 

Micellar NP formulations have revolutionized transdermal therapeutics [43], enabling high concentrations of drugs to permeate the skin, forming a drug depot in the epidermis [44]. This route of administration minimizes gastrointestinal contact and hepatocyte’s first-pass effects, and it is more cosmetically tolerable to patients [42]. Physiochemical characteristics of micellar NPs formulations can be modified for various routes of administration [45], e.g., by changing miscibilities to optimize transdermal penetration, combining a mucoadhesive for vaginal administration, changing the particulate size, or adjusting the zeta potential versus in suspension [42,46]. The following are illustrative examples of micellular formulations.

Polymeric micelles from hyaluronic acid (HA) of 200 nm did not stay on the epidermis but permeates it. Optical microscopy images revealed HA was in keratinocytes and fibroblasts and micelle disruption begins at approximately 10 mm deep in the skin [47]. 

Vinpocetine (VPC)-loaded D-α-tocopherol polyethylene glycol 1000 succinate (TPGS) and alpha lipoic acid (ALA) (VPC-TPGS-ALA) film was used for transdermal drug delivery [48]. This film enhanced VNP penetrability in the epidermis compared to raw VNP-loaded transdermal film; optimized TPGS-ALA gave improved penetration in the epidermis after 0.5, 2.0, and 4.0 h compared to the raw VNP-loaded transdermal film [48]. 

Azole antifungal agents-loaded micellar NPs increased in skin which led to deposition in hair follicles as revealed by confocal microscopy. Skin absorption using methoxy-poly(ethylene glycol)-di-hexyl-substituted lactide (MPEG-dihexPLA) micelles can enhance epidermal drug bioavailability, which might result in increased efficacy of in vivo treatment [49].

### 2.2. Magnetic NPs

Super paramagnet magnets (Fe_3_O_4_) and magnetite (Fe_2_O_3ɣ_), commonly used in medical applications [50], have particle sizes of 3–30 nm and size distributions of ~10–20% and they dissolve in water [51,52]. Iron oxide (Fe_3_O_4_) magnetic NPs of a small size are favored for biomedical and biological applications [53].

Magnetic NPs used in pharmaceutical applications have small sizes [54], good biocompatibilities [55,56], are easy to process [57], and can be endowed with unusual characteristics [58]. The number of research papers on magnetic NPs has dramatically increased over the last two decades [59,60,61,62,63,64]. Their magnetostrictive responses enable biomolecules to be magnetically detected, allowing new exciting perspectives on bio-separation [65,66], bio-detection [67,68], and targeted drug administration [69]. Furthermore, external magnetic fields can heat them, providing treatment options via magnetic fluid hyperthermia [70]. 

In one illustrative study [71], Fe_3_O_4_ NPs were used as a core coating, followed by creating laser-sensitized magnetic nanoparticles (LMNs) loaded bacteria cellulose membrane (LMN/BC). Furthermore, the results reveal that laser-activatable and magnetostrictive LMNs synergistically impact breast cancer inhibition, making it an alternative treatment option for superficial cancer [71].

A recent report described how cobalt ferrite magnetic (CoFe_2_O_4_) NP could be used to remove azithromycin, an antibiotic for treating infectious diseases such as COVID-19, from hospital effluents via UV illumination [72]. CoFe_2_O_4_ NPs are attracted to magnetic fields and generate heat when they move; this magnetic fluid hyperthermia is central to these noninvasive cancer treatments [73]. Similarly, nickel (Ni) NP have a high surface area relative to their size, allowing for a large capacity of drugs to be carried in the drug delivery system, imaging, and cancer treatment [74].

### 2.3. Hollow NPs

Hollow NPs with penetrable and porous shells have unique properties relative to solid ones: a high surface area, high loading capacity [75], less expensive [76], and low density [77]. Research shows that hollow NPs with a high surface area could effectively accommodate strain throughout a chemical reaction while promoting complete electrolyte penetration [78]. The following are illustrative studies featuring hollow NPs.

Semiconductor hollow copper sulfide nanoparticles (HCuSNPs) are synthetic photo-absorbers [79] with a diameter ranging from 3 to 11 nm [80]. Furthermore, the photo-thermal [76,81] features of the HCuSNPs enable them to capture near-infrared light and reach temperatures of around 40 °C before partially damaging the stratum corneum and allowing the drug to deliver safely [76]. Thus, HCuSNPs have been used to deliver chlorine6 (Ce6, a photosensitizer) and doxorubicin (DOX) to the 4T1 mouse mammary cells [82]. The thermo-responsive degradation of HCuSNPs can trap drugs for controlled release via light-induced thermal stimuli. This resulted in the minimal clearance of drugs non-specifically in the circulation, thereby increasing drug bioavailability in tumor tissues by improving permeability and retention effects [82]. 

Recently, Zan et al., 2022, incorporated copper sulfide nanodots (CuSND) into the intravenous thermo-responsive hydrogel to elicit alleviating remodeling via transdermal mild photo-thermal therapy on the transdermal white adipose depot. Meanwhile, mirabegron was also co-administered with CuSND hydrogels, resulting in a substantial therapeutic synergistic effect. The result demonstrated that in vivo CuSND hydrogel treated with high-fat diet mice were found to have lower triglyceride serum levels, insulin, cholesterol, and glucose, and enhanced insulin sensitivity compared to the untreated group [83]. 

Another study shows reactive oxygen species (ROS)-responsive hollow mesoporous silica nanoparticles (HMSNs) loaded with Glabridin were investigated by [84] as a nanocomposite for transdermal drug delivery and anti-pigmentation. The results demonstrated that HMSN-CD/1-adamantanemethylamine tagged polyarginine peptides (Ada-R8) (HMSN-CD/Ada-R8) loaded with Glabridin improved the broadening of Glabridin usage and achieved a great photoprotection ability. The NPs were highly stable in aqueous solutions and demonstrated good biocompatibility while releasing Glabridin in a controlled ROS-responsive way [84].

In previous study, (Wang et al., 2021) has produced glucose-responsive polymer grafted hollow mesoporous silica (HMSNs-PAPBA) NPs for in vivo culture for diabetes Sprague Dawley (SD) rats. The results show that drug diffusion was effectively inhibited and fast released under typical hyperglycemia conditions in an average blood glucose level [85]. 

Likewise, Zhang et al., 2018, developed the polydopamine/lauric-acid-coated (PDA/LA-coated) hollow mesoporous SiO_2_ for transdermal delivery. The result showed that the PDA/LA-coated hollow mesoporous SiO_2_ exhibits a good photothermal-response, and is non-toxic in in vivo type II diabetes SD rats [86].

### 2.4. Hydrogel NPs

Hydrogels have a high water absorbing capacity [87,88] which can be integrated as drug carriers for transdermal drug delivery [88] due to their adhesion ability to the skin surface [89]. The porosity of the semisolid morphology of a hydrogel matrix allows a higher rate of drug loading and release [90,91]. 

Hydrogels can be biocompatible and biodegradable, particularly if made from natural materials. They are often used as biopolymers in transdermal drug delivery due to their hydrophilic properties and sensitivity to external stimuli [90,92]. Transdermal drug delivery using hydrogels has unique features including prolonged release behavior [89,93,94]], low toxicity [95], and liver damage prevention [89]. Table 1 summarizes the recent studies of hydrogel in transdermal drug delivery.

### 2.5. Poloxamer Hydrogels

Poloxamers hydrogels are thermosensitive polymers commonly employed in situ [101]. Poloxamer 407 (P407) is widely used in transdermal applications due to its good absorbing ability [102], prolonged drug release [101], low toxicity [103], and high biocompatibility [104]. P407 hydrogels have been studied as potential nanocarriers in drug delivery systems owing to their non-irritating action on cellular membranes and sustained release of drugs with minimal side effects [105,106]. 

According to transdermal research, a P407 hydrogel matrix loaded with carboxymethyl cellulose sodium (CMCs) has been developed to investigate the transdermal permeability in porcine ear skin [107]. As observed in the study, CMCs improved the porosity of a P407 hydrogel structure, and the developed P407/CMCs hydrogel enhanced the overall drug penetrability into porcine ear skin without using chemical effectors [107]. P407/CMCs hydrogels have also been developed to treat atopic dermatitis (AD) and the results showed that the hydrogel exhibited favorable percutaneous [108]. The FESEM images showed that the porosity of the P407/CMCs hydrogel was increased due to the presence of CMCs, which consequently facilitated the drug release across the skin, as shown in Figure 1 [107].

### 2.6. Acrylic Acid Copolymer Hydrogels

Furthermore, VACPH hydrogel was produced by copolymerizing acrylic acid (AA), vinyl benzyl trimethylammonium chloride (VBTMACl), polyvinylpyrrolidone (PVP), and choline ionic liquid (ChMACl) [109]. The strong polarization of ammonium cations in VBTMACl means microwaves enhance their thermal conversion capabilities, resulting in the heat killing of *S. aureus* and methicillin-resistant *Staphylococcus aureus* (MRSA) [109]. In another application, fluconazole and functionalized gold nanorods were conjugated and incorporated into a P407 hydrogel to produce a nano complex structure for non-toxic transdermal delivery to human dermal fibroblasts CCD-1064Sk cells [110].

Polyacrylamide (PAAm)-grafted-pectin (PCT) (PAAm-g-PCT) hydrogel was produced by free radical polymerization and alkaline hydrolysis techniques as rate-controlling membranes (RCMs) in an electro-sensitive transdermal drug delivery system [111]. The result reveal that drug permeability reduced with an increase in glutaraldehyde concentration and drugs in electro-sensitive transdermal drug delivery systems increased with an applied electric stimuli. The same group produced a polyacrylamide–graft–pullulan (PAAm-g-PLN) hydrogel and showed that drug permeation was minimal when no electric stimulus was applied but diffusion rates increased with electric stimulation [112]. 

## 3. Poly(lactide-*co*-glycolide) (PLGA) NPs

PLGA NPs [113,114] are particularly important due to their biodegradability [115], biocompatibility [116], lack of toxicity [117], and capacity to protect biomolecules from degradation [114]. PLGA NP sizes vary from 100 to 5000 nm [118] and the average intracellular delivery range is from 107.7 nm to 245.7 nm [119]. They hydrolyze in the body to produce innocuous smaller synthetic polymeric materials comprising lactic and glycolic acids [117,120], Figure 2.

PLGA NPs are attractive for drug delivery and tissue engineering. Size of PLGA NPs can be tailored to suit particular drug delivery applications [121]. Smaller nanoparticles (<100 nm) deliver drugs to cells or tissues [122], while larger nanoparticles (>500 nm) deliver drugs to targeted organs or the entire body [123]. According to one reference [124], nanoprecipitation of PLGA NPs alters their sizes, causing increased intracellular delivery.

PLGA NPs can be functionalized using agents targeting moieties [125,126,127] or polyethylene glycol (PEG) [128,129,130] to improve stabilities. According to [131], PEGylated PLGA NPs and folate-functionalized PLGA NPs have been used to deliver chemotherapy drugs to targeted cancer cells. Moreover, the PLGA NPs are biodegradable in the body [132]; this is useful for sustained release, eg to treat glaucoma [133,134] or osteoarthritis [135,136,137].

The skin permeability of rat abdomen skin via indomethacin (IM)-loaded PGLA NPs was studied using antisolvent diffusion. It revealed that skin penetration indomethacin of 50 nm and 100 nm PLGA NPs with iontophoresis was substantially higher after 2 h administration. The use of iontophoresis resulted in a greater penetration of 50 nm PLGA nanoparticles into rat skin as compared to 100 nm PLGA nanoparticles [138]. The same group of researchers developed 17 β-estradiol (E2)-loaded PGLA NPs. The results show higher density skin permeability in E2-loaded PGLA NPs than polyvinyl alcohol (PVA)-coated NPs; it also enhanced bone mineral density of the cancellous bone in an osteoporosis animal model [139]. Both studies conclude that transdermal delivery of PLGA NPs combined with IP will deliver the drug deep into the rat hair follicles [140]. Table 2 summarizes the recent studies of PGLA in transdermal drug delivery.

### 3.1. Ethylene-Vinyl Acetate Flims

Solvent casting has been used to incorporate selegiline hydrochloride (SGN)-loaded PLGA NPs in ethylene-vinyl acetate (EVA) transdermal film. Field emission scanning electron microscopy (FE-SEM) analyses showed these NPs were smooth spherical surfaces. In rat’s brain tissues, a small dose of reserpine initiates symptoms of Parkinson’s disease by increasing monoamine oxidase B (MAO-B) level, and decreasing dopamine [141].

### 3.2. Poly(lactic-co-glycolic acid) Coating

The same group of researchers created PLGA-coated rasagiline mesylate-nanoparticles (RM-NPs) and loaded gellan gum in the transdermal film via solvent evaporation and a solvent casting method. They reported that gellan gum with 1.127 g transdermal hydrogel aids in non-chronic drug self-administration for >72 h without skin inflammation [142].

**Table 2 biomedicines-11-01124-t002:** Recent study of PLGA in transdermal drug delivery.

Drug Delivery System	Method	Application	Result	References
PGLA/collagen scaffold	Electrospinning	Human dermal fibroblast and human keratinocyte	High mechanical strength, good surface adhesion on both cell lines	[143]
Poly(dl-lactide-*co*-glycolide)-poly(ethylene glycol)-poly(dl-lactide-*co*-glycolide) copolymers (PLGA-PEG-PLGA) NPs	Antisolvent diffusion method	Rat skin	High thermodynamic activity, skin permeability and low irritation in PLGA-PEG-PLGA NPs	[144]
Gentamicin loaded PLGA (GM-PLGA) NPs	Solvent evaporation method	Rabbit	No sign of inflammation and non-toxic to all groups of rabbit	[145]
Hyaluronate-PGLA (HA-PGLA) NPs	Solvent evaporation method	Rat skin	No cytotoxicity, biocompatibility in cell viability, and high efficiency of transdermal delivery	[146]
Dictamnine-PGLA-nanocarrier (Dic-PGLA-NC)	Ultrasonication	Mouse dermatitis model	Dic-PGLA-NC can penetrate the dermal layer effectively and achieve sustained drug release	[147]

## 4. Chitosan NPs

Chitosan is a natural polymer that emerged as one of the useful nanocarriers for various therapeutic agents in transdermal drug delivery. Functional properties such as biocompatibility and biodegradability [148] have facilitated the development of chitosan-based nanocarriers for medicinal applications [149,150]. Slow polymer erosion in chitosan allows effective drug encapsulation [151] for sustained and controlled drug release with low toxicity. For example, the functional properties (amine and hydroxyl functional groups) of chitosan can be altered by adding synthetic materials to fabricate microneedles for transdermal applications as it does not create an unwanted immune response in the body. Therefore, chitosan is notable for its role as a nanocarrier of hydrophilic therapeutic drugs [152].

### 4.1. Chitosan-Sodium Alginate

In research carried out by [153], chitosan-sodium alginate (CHI-SA) nanogel was synthesized to perform the transdermal delivery of the drug pirfenidone to treat pulmonary fibrosis. According to their observation, the loading capacity of the nanogel was ≈50% and the encapsulation efficiency was as high. The permeation of the drug pirfenidone across the skin has been remarkably enhanced using CHI-SA nanoparticles in an ex vivo study. In any case, the in vitro drug release profile has demonstrated the burst release of the drug pirfenidone in the first 5–7 h (≈12%) followed by sustained drug release behavior [153].

### 4.2. Chitosan-Chondroitin

An antimalarial agent called artemether has been encapsulated into chitosan-chondroitin sulfate nanoparticles and subsequently loaded into transdermal patches to treat acute malaria, as reported by [154]. The positively charged chitosan were interacting with negatively charged chondroitin sulfate through the ionic gelation method to form nanoparticles that offer less toxicity, high stability, encapsulation efficiency, and loading capacity for transdermal delivery. The ex vivo study showed that using olive oil increased the drug permeability of the transdermal patch as a permeation enhancer. Moreover, a high cumulative drug release at a pH of 7.4 in contrast to a slow drug release rate at a pH of 5.5 was reported in their in vitro drug release study.

### 4.3. Chitosan-Nanoelmusion Films

Nanoemulsions have proven to enhance the transport of drug molecules in transdermal analgesic patch systems. For instance, da Silva et al., 2020, have incorporated nanoemulsions into chitosan films to carry methyl salicylate, which displayed a homogenous formulation with no phase separation. The chitosan-nanoemulsions films were reported to have a higher loading capacity of methyl salicylate and moisture levela as compared to physical mixture films [155]. In another study, chitosan (CS) and polyvinyl alcohol (PVA) was combined and crosslinked with vanillin (VA) to form a matrix transdermal system (Figure 3) to deliver the enrofloxacin drug, as reported by [156]. From their research findings, CS-PVA films can be produced by a simple solvent-casting method and the CS-PVA-VA nanocomposite has demonstrated sustained enrofloxacin release in the in vitro drug release profiles. The released kinetic study has confirmed that by altering the vanillin concentration, the diffusion mechanism and drug release rate can be controlled. As reported, the drug loading capacity was successfully increased by crosslinking 3% vanillin with the CS-PVA films. The controlled release of the enrofloxacin drug was achieved due to the lower swelling ratio of the 3% vanillin crosslinked films which also reduced the initial burst release [156].

### 4.4. Chitosan-Coated Lipid Carriers

In previous research [157], chitosan-coated nano lipid carriers (Ch-NLC) were used in the transdermal delivery of tetrahydrocurcumin (THC) to treat triple-negative breast cancer (MD-MBA-231). THC-Ch-NLC is proven to demonstrate high skin permeation through an artificial membrane and enhanced cellular uptake with significant cytotoxicity to the MD-MBA-231 breast cancer cells. The THC-Ch-NLC also showed the sustained release of the drug through an in vitro release study which was confirmed by the Korsmeyer–Peppas model [157]. In another interesting study carried out by [158], nanoparticles made from chitosan whisker (CSWK) with oligo(lactic acid) (OLA) were used to transport lidocaine in the reticular dermis. They claimed that the CSWK-OLA nanocarriers offered high drug permeability into the skin without penetration enhancers and active strategies such as microwave technology and iontophoresis. It is due to the nano size of CSWL-OLA particles with amphiphilic properties, allowing the lidocaine-loaded nanocarriers to penetrate deep into the reticular dermis [158]. 

Table 3 below summarizes the incorporation of chitosan with different nanocomposites in various transdermal delivery systems.

## 5. Carbon Nanotubes CNTs

These have gained enormous research attention in the past decades for drug delivery applications, but there have been only applications in transdermal drug delivery. This may be due to poor skin penetration ability without the application of mechanical stress. Therefore, functionalized CNTs (f-CNTs) have been developed for greater biocompatibility and enhanced transdermal properties.

### 5.1. Functionalized Multi-Walled Carbon Nanotubes 

The f-MWCNTs-KP-ES nanocomposite was reported high in drug loading rate, stability, and encapsulation efficiency as compared to CNTs or ES alone. Ex vivo results showed a fast skin penetration rate. Anyway, the f-MWCNTs-KP-ES nanocarriers had successfully prolonged the release of ketoprofen *in vivo* and caused no adverse effect on the observed rat skin [172]. 

### 5.2. Controllable CNT Membranes

A controllable CNT membrane device has been created for the transdermal delivery of nicotine in guinea pigs [173,174]. This CNT membrane device had successfully delivered nicotine fluxes when switched ON and OFF, respectively, according to an in vitro flow-cell test; meanwhile, nicotine flux has been detected through in vivo microdialysis with a membrane implanted in the guinea pig’s skin. Nevertheless, the CNT membrane has low flexibility and tends to flatten the skin contact area which requires applying a small amount of hydroxyethyl cellulose gel on the skin surface underneath the membrane device [173,174].

### 5.3. “Bucky Paper”

Carbon nanotube film, also known as “buckypaper”, has been proposed for use in actuating, structural, and filtration systems, owing to their distinct and robust mechanical properties [175]. Bucky papers [176] assembled into functionalized single-walled CNTs (f-SWCNTs) and multi-walled CNTs (f-MWCNTs) in the transdermal delivery of four model drugs namely clonidine hydrochloride (CHC), selegiline hydrochloride (SHC), flurbiprofen (FB), and ketorolac tromethamine (KT). According to the in vitro transdermal test carried out in the study, using electrical bias can control the drug release rate and direction, where polarities rely on the charge of the drug. CNTs–bucky papers loaded with drugs demonstrated passive drug release behaviors which are highly dependent on the types of drugs used and the electromodulation was successfully applied to accelerate and decelerate the drug release rate [176].

### 5.4. CNT Gold NPs 

Another similar study using electro-permeabilization was reported by [177], where CNT was incorporated with gold nanoparticles (GNP) into a transdermal patch matrix of polyvinyl alcohol/poly(dimethyl siloxane)-g-polyacrylate to deliver the drug diclofenac sodium (DS). The results showed a significant transdermal effect from the 1.5%-GNP-CNT patch at an applied bias of 10 V with no initial burst release. The 1.5%-GNP-CNT patch had the highest drug encapsulation efficiency, as compared to the other patches. Overloading of GNP-CNT will decrease the drug encapsulation efficiency remarkably due to the particle’s agglomeration. The cell viability assay on HaCaT cell lines showed no cytotoxicity to the cells, suggesting good biocompatibility when applied to human skin [177].

### 5.5. CNT Hydrogel Hybrid

Another group of researchers suggested embedding double-walled carbon nanotubes (DWCNTs) into hydrogel for potential transdermal drug delivery applications through electro-permeabilization [178]. In their study, hydrogel nanocomposites combining agarose (AG) and DWCNTs were prepared at different concentrations (DWCNTs-AG) and an agarose hydrogel without DWCNTs was used as the control (CTRL-AG). Due to the higher concentrations of DWCNTs in the hydrogels that shorten the electron jumping distance across the applied electric field, their results show the DWCNTs-AG nanocomposite had higher conductivity across the applied electric field as compared to the CTRL-AG nanocomposite, having the potential to increase skin permeability for transdermal delivery. Nevertheless, there was no show of in vitro and in vivo tests in the same study to verify skin permeability results [178]. 

## 6. Nanocellulose NPs

Nanocellulose (NC) is a cellulosic polysaccharide [179] widely used in antimicrobial applications in various industries. This material can be formulated into drug carriers [180] for transdermal drug delivery [181], and wound dressings. It has a large surface area [182], but remains light-weighted, biodegradable [183], biocompatible [182,184], and has a low production cost [185]. NC can combine with other biopolymers such as chitosan to enhance the structural properties of the composite structure due to its high tensile strength [186,187].

NC transdermal drug delivery is favored because it provides pain-free application [188,189,190], high water permeation [191], prolonged response [192,193], and therapeutic effectiveness in a low amount of dosages [194,195].

### 6.1. Bacteria Nanocellulose

Bacterial nanocellulose (BNC) is a natural biomaterial [196] with unique characteristics, including low toxicity [197], biocompatibility [198], high purity [199], and a nano-porous structure. BNC-based products have been approved by the Food and Drug Administration (FDA) and Conformité Européenne (CE) for biomedical applications [200]. Numerous studies found that BNC membranes can be effectively loaded with diverse biological molecules with varying permeability and hydrophilicity [201]. Moreover, BNC membranes have previously been combined with drugs and bioactive compounds, including lidocaine [202], ibuprofen [203], and amoxicillin [204], or developed as ionic liquids [205,206] for transdermal drug delivery.

The recent work proposed by [207] was to develop patches using hyaluronic acid (HA), microneedles (MNs), and bacterial nanocellulose (BC) (HA-BC MNs). HA was utilized as a biomacromolecule with moisturizing, regenerative, and hydrating properties, while BC was used to protect the inclusion of another bioactive molecule such as rutin to illustrate the system’s efficacy. The HA-BC MNs patches exhibit appropriate morphology, mechanical resistance, and biocompatibility. The in vivo patches applied on human volunteers show tolerability as a dermo-cosmetic system and open up new opportunities for the incorporation of different active ingredients to broaden their application [207].

Another study (Abba et al., 2019) produced a crocin-BNC membrane for transdermal drug delivery. The swelling analyses and resonance peaks of FTIR analysis revealed that BNC had good crocin uptake. The crocin-BNC surface morphology indicated that the fibers were unchanged, with no fiber damage. The Franz diffusion tests showed that drug permeability was steady and long-lasting [208]. A nanocomposite film of bacterial cellulose (BC) modified with dopamine (DPM) and incorporating reduced graphene oxide (rGO)/silver (Ag) (BC-DOPA/rGO/Ag) NPs was successfully produced by [209] for antimicrobial patches. The result demonstrated that Ag NPs in the nanocomposite promote cell development and migration in NIH 3T3 fibroblast cells and A549 human lung epithelial cells, resulting in a faster wound-healing process. Another recent research study discovered that incorporating Fe_3_O_4_ NPs coated with DOX on BC could be used in the treatment of breast cancer because they can permeate through the epidermis under magnetic flux and laser radioactivity [71] 

Recent research investigated the depot stability of BNC membranes loaded with various APIs such as caffeine, lidocaine, ibuprofen, and diclofenac, and showed all systems were stable, with no morphological changes or differences in the drug release profile under optimal storage conditions. Moreover, the caffeine-loaded BNC membrane was selected for in vivo epidermal studies and the results revealed the APIs-loaded BNC membranes’ good storage stability [201]. 

### 6.2. Cellulose Nanofibers

Cellulose nanofibers (CNFs) provide large areas for drug–CNF interaction and mechanical characteristics that improve the dosage from mechanical stability [210]. Moreover, CNF films have excellent water vapor permeability properties at low humidity [211], which improves the storage stability of oxygen-sensitive drugs during storage and allows them to be used effectively as excipients [193,212]. Recent CNF research focuses on producing environmentally sustainable nanocomposites which do not exhibit any adverse outcomes associated with the synthetic nanomaterials widely used for reinforcements [213,214].

#### 6.2.1. With Poly(N-isopropyl acrylamide)-Graft-Guar Gum (GG-g-PNIPAAm)

In another study [215], CNF produced from jute fiber was reinforced with a poly(N-isopropyl acrylamide)–graft–guar gum nanocomposite to test the controlled release of diltiazem hydrochloride in transdermal drug delivery. Compared to GG-g-PNIPAAm, the nanocomposite films demonstrated greater thermal resistance and barrier properties. The nanocomposite film containing 1wt% CNF performed better than other films. The toxicity test confirmed that the GG-g-PNIPAAm nanocomposite film is non-toxic to rat skin. Furthermore, the ideal nanocomposite’s in-vitro release study revealed a controlled diltiazem hydrochloride release capability. As a result, GG-g-PNIPAAm contains 1wt% CNF nanocomposite that can be used as a transdermal patch due to its outstanding physicomechanical, bio-interfacial, as well as permeability properties.

#### 6.2.2. With CNF Transdermal Films

In a previous study, an electrospinning method was used to produce polyurethane/hydroxypropyl cellulose (PU/HPC) electrospun nanofibers [216]. In vitro studies across the skin imitating a polymeric membrane demonstrated model drug flux. The MTT assay showed the PU/HPC electrospun nanofiber non-toxic to mouse fibroblast cell line (BALB/3T3). These findings revealed that an PU/HPC electrospun nanofiber could be used as a transdermal drug delivery system [216]. Recent research isolated CNF from jute fibers and nano-collagen (NCG) from waste fish scales to form a CNF-NCG biocomposite by the electrospinning method to test the sustained release of ketorolac tromethamine (KT), as shown in Figure 4. In vitro drug release results revealed that the CNF-NCG (1wt%)-loaded polyvinyl alcohol/methylcellulose/polyethylene glycol (PVA/MC/PEG) bio-nanocomposite demonstrated an excellent sustained drug release of ketorolac tromethamine, making it an ideal biocomposite for transdermal drug delivery systems [217]. Another study (Sarkar et al., 2017) proposed using a CNF/chitosan transdermal film to deliver ketorolac tromethamine. According to the release profile, researchers discovered that CNFs effectively resulted in prolonged drug release. The XRD analysis revealed the rise in crystallinity with the addition of CNFs in CNF/chitosan transdermal films [218].

### 6.3. Cellulose Nanocrystal

Cellulose nanocrystals (CNCs) are promising due to their remarkable properties such as biocompatibility [219], non-toxicity [220], and high mechanical strength [221]. It is also utilized in transdermal preparations to improve clinical outcomes and the possibilities and potential benefits of developing various management strategies in interdisciplinary research [222].

#### 6.3.1. With Methylcellulose

In a recent study, CNCs derived from jute fibers were used to develop a non-toxic bio-nanocomposites transdermal patch formulated with methylcellulose (MC) and chitosan (CH) via the solvent evaporation method to analyze the sustained drug release of ketorolac tromethamine [223]. The results showed that adding CNC nanofillers in MC/CH improved the thermal properties of the bionanocomposite patches. Furthermore, incorporating CNCs into the MCCH blend improved the permeability, water absorption, mechanical properties, and sustained drug release ability than the pure matrix. The toxicity assay of the formulated bio-nanocomposite (MCCH1) revealed that it has a low cytotoxicity and performed good outcomes for transdermal drug delivery systems [223].

#### 6.3.2. CNC-Hydrogels

The most recent method proposed by [224] used inversion and tilting methods to produce a crosslinked CNC/donepezil hydrochloride (cCNC/DPZ) hydrogel to evaluate long-acting drug delivery through subcutaneous injection and reported that the addition of DPZ to the CNC dispersion caused gel aggregation, and pH control of the CNC/DPZ hydrogel increased elastic modulus. In any case, the cCNC/DPZ hydrogel showed a longer half-life, high mean residence time, and lower C_max_ values compared to the DPZ and CNC/DPZ hydrogel in a pharmacokinetic study [224].

#### 6.3.3. Lanoconazole (LCZ)-Loaded CNC

LCZ-loaded CNC grafted with polyphosphoesters (LCZ-loaded CNC-PEs) were produced by [225] to improve the chronic inflammation efficiency of LCZ on mouse ear skin. The result revealed a high LCZ efficiency and small mean droplet size in CP-PEs. The long-lasting local action was assured by the sustained LCZ release and better transdermal delivery of the LCZ-loaded CP-PEs, probably due to the oil droplets’ rigidity. Furthermore, a mouse ear model of 12-O-tetradecanoylphorbol-13-acetate (TPA)-induced inflammation showed excellent anti-inflammatory efficacy of the LCZ-loaded CP-PEs [225]. 

## 7. Ionic Liquids (ILs)

Ionic Liquids (ILs) are organic salts comprised of an organic cation and an organic or inorganic anion that, when mixed in 1:1 molar ratio (true ionic liquid), give rise to a room-temperature ionic liquid (RTIL). Ionic liquids can solubilize amphipathic molecules and increase drug solubility and can favor topical drug delivery. The ionic liquid molecules are likely to slip through the fatty compounds that make up skin cells, creating small transient openings through which bioactive molecules (carried by ionic liquid) can permeate. 

### 7.1. Choline Geranic Acid (CAGE)

In particular, choline and geranic acid (CAGE) has been used to enhance the transdermal delivery of several small and large molecules including proteins such as bovine serum albumin (BSA, molecular weight: ≈66 kDa), ovalbumin (OVA, molecular weight: ≈45 kDa) as well as insulin (INS, molecular weight: 5.8 kDa) [226]. 

Two major applications of CAGE are biofilm-disruption and enhanced antibiotic delivery across skin layers. Relatively few papers describe both these applications concomitantly, but the first was the use of neat ILs as antimicrobial agents and transdermal drug-delivery agents [227]. In another study, CAGE increased the delivery of cefadroxil, an antibiotic, by >16-fold into the deep tissue layers of the skin and is able to induce >95% bacterial death after a 2 h treatment. Further research on CAGE revealed that it exhibits broad-spectrum antimicrobial activity against several drug-resistant bacteria, fungi, and viruses including clinical isolates of *Mycobacterium tuberculosis*, *Staphylococcus aureus*, and *Candida albicans* and laboratory strains of Herpes Simplex Virus. CAGE affords negligible local or systemic toxicity, and an approximately 180–14,000-fold improved efficacy/toxicity ratio over the currently used antiseptic agents in human keratinocytes and mice studies. CAGE penetrates deep into the dermis and treats pathogens located in deep skin layers. Thus, CAGE have been used in vivo to treat *Propionibacterium acne*. Overall, these studies demonstrate the promise of CAGE as transformative platforms for antiseptic agents used prophylactically as well as therapeutically [228]. 

Low percentage-loaded CAGE-IL (viz. 2.0%, *w*/*w*) are effective for facilitating the passage of curcumin to transiently disrupt the skin barrier [229]. It emerged that the anion–cation ratio is fundamental in the design of suitable ionic liquids. This parameter can significantly alter their physical properties and interactions with biological tissues. Transport enhancement is also composition-dependent, since when different ratios of CAGE: 1:1, 1:2, 1:4, and 2:1 components exhibited variable dermal insulin delivery [230]. Similarly, choline oleate ionic liquid-based CAGE-promoted transdermal insulin permeation depends on the choline geranate rations, with 1:2 being the best found (better efficacy with less cyto- and geno-toxicity [231].

### 7.2. Surface Active Ionic Liquid (SAIL) 

Besides that, IL has been used in formulations of nano-drug delivery system. Researchers from Japan developed ionic liquid (IL)-in-oil microemulsion formulations (MEFs) for transdermal insulin delivery using choline-fatty acids ([Chl][FAs])-comprising three different FAs (C18:0, C18:1, and C18:2) for biocompatibility). The MEFs were developed using [Chl][FAs] as surfactants, sorbitan monolaurate (Span-20) as a cosurfactant, choline propionate IL as an internal polar phase, and isopropyl myristate as a continuous oil phase. MEFs significantly enhanced the transdermal permeation of insulin via the intercellular route by compromising the tight lamellar structure of SC lipids through a fluidity-enhancing mechanism. The in vivo transdermal administration of low insulin doses (50 IU/kg) to diabetic mice showed that MEFs reduced blood glucose levels (BGLs) significantly compared with a commercial surfactant-based formulation by increasing the bioavailability of insulin in systemic circulation and sustained the insulin level for a much longer period (half-life > 24 h) than subcutaneous injection (half-life 1.32 h) [232]. 

Another research study reported an advantageous carrier for the transdermal delivery of paclitaxel (PTX) comprising a new micelle formulation (MF) that consists of two biocompatible surfactants: cholinium oleate ([Cho][Ole]), which is a surface-active ionic liquid (SAIL), and sorbitan monolaurate (Span-20). A solubility assessment confirmed that PTX was readily solubilized in the SAIL-based micelles via multipoint hydrogen bonding and cation–pi and pi–pi interactions between PTX and SAIL[Cho][Ole] [233]. A similar group of researchers developed a protein-containing nanocarrier (PCNC) comprising an antigenic protein (ovalbumin/OVA) stabilized by a combination of surfactants, i.e., a lipid-based surface-active ionic liquid (SAIL) and Tween-80. The PCNC was biocompatible both in vitro and in vivo, and is suitable for use in therapeutic transdermal drug delivery. The skin permeability of the PCNC was significantly (*p* < 0.0001) enhanced, and the transdermal distribution and transdermal flux of the OVA delivery system were 25 and 28 times greater, respectively, than those of its aqueous formulation. The PCNC disrupted the order of lipid orientation in the skin’s SC and increased intercellular protein delivery. It demonstrated effective antitumor activity, drastically (*p* < 0.001) suppressed tumor growth, increased mouse survival rates, and significantly (*p* < 0.001) stimulated the OVA-specific tumor immune response. The PCNC also increased the number of cytotoxic T cells expressing CD8 antibodies on their surfaces (CD8 + T-cells) in the tumor microenvironment. These findings suggest that PCNCs may be promising biocompatible carriers for transdermal antigenic protein delivery in tumor immunotherapy [234].

## 8. Natural Rubbers

Rubber can be natural from rubber trees [235], or synthetic from petroleum byproducts [236,237]. Natural rubbers are widely used in biomedical industries [238] and transdermal drug delivery systems [239] due to their biocompatibility [240], excellent mechanical properties [241], flexibility, and ability to form films easily [242]. It is a colloidal suspension composed of particles with poly(cis-1,4-isoprene) chains [238]. The most common natural rubber-based transdermal patches are nanocomposites, which serve as the polymer matrix, and the properties of transdermal patches depend on the type of penetration enhancers used [237]. 

### Natural Rubber Layers

A previous work by Marcelino et al., 2018, used the casting method to produce a fluconazole-loaded NRL to examine the *Candida albicans* (*C. albicans*) antifungal susceptibility. The release of fluconazole inhibited the growth of *C. albicans* for 48 h, indicating good properties for use as transdermal patches. Furthermore, adding fluconazole to NRL did not significantly alter the mechanical properties of the latex, resulting in a promising biomaterial for transdermal applications [243]. On the other hand, [244] used the casting method to produce a voriconazole–NRL (VCZ-NRL) membrane to evaluate the antifungal susceptibility of *Candida parapsilosis* (*C. parapsilosis*)-infected ulcers, and the results revealed no hemolytic effects or mechanical adaptability for dermal application. Anyway, the VCZ was released in two stages: a burst release of 13.2% of an initially incorporated VCZ in 1 h, followed by a slow release of 11% VCZ up to 48 h. The VCZ-NRL membrane performed well in mechanical, antifungal, and physiochemical tests, making it an intriguing alternative to treating Candida-infected wounds [244]. In addition, [245] discovered ketoprofen–NRL membranes for the treatment of tendinitis. The results showed that biocompatible NRL membranes demonstrated a 60% sustained ketoprofen release in 50 h. Furthermore, adding ketoprofen into the NRL membrane was non-toxic to red blood cells [245]. 

Poly(p-phenylenevinylene) (PPV) was incorporated into a natural rubber latex (PPV-NRL) composite for electromechanical transdermal drug delivery [246]. Ibuprofen was used as doped for PPV, which acted as the drug polymeric carrier. The findings indicate that ibuprofen-NRL patches and ibuprofen-doped PPV/NRL matrices were successfully produced under ultraviolet (UV) radiation at various crosslinking metrics. In any case, using an electric flux will improve the penetration of drugs in the synergic effect by developing the PPV chain, changing the conductive polymer’s oxidation condition, and the intravenous administration impact, as well as the continuation of pore size in keratinocytes. The NRL patches had a higher capacity modulus value and better physicochemical characteristics than transdermal drug delivery patches made from hydrogels.

## 9. Conclusions

NP-based carriers are a safe and efficient media for transdermal drug delivery. NP carriers can have high stabilities, reduced toxicities, biodegradability, high loading capacities, good storage stabilities, and can also be incorporated with hydrophilic and hydrophobic materials for controlled, time-dependent drug delivery [247]. They have provided viable treatment options for a variety of diseases, particularly to deliver drugs to cancer legions. For example, NPs are extensively studied as potential carriers to transport drugs to the lining of blood vessels to treat arteriosclerosis and myocardial infarction [248,249]. At present, the most promising NP-based carriers are from polylactic acid (PLA-NP) or polyethylene glycol (PEG-NP). These have been used for controlled drug delivery to the liver and brain, and as multi stimuli-responsive entities to treat inflammation in rheumatoid arthritis or asthma. However, the slower biodegradation of polymeric NP carriers is a “double-edged sword” with regards to the chronic accumulation of toxic metabolites associated with causes of systemic toxicity.

## 10. Future Perspective

Optimal particle sizes for NP-based drug delivery systems could be further investigated as they are crucial in drug distribution, release rates, targeting efficacies, and toxicities; current understandings are probably overgeneralized as follows. Relationships between particle sizes and skin permeability are such that nanovesicles with 70 nm diameter or less are effective in delivering contents to epidermal and dermal layers, as compared to those with a diameter of 300 nm [250]. On the other hand, nanovesicles 600 nm diameter or more tend to remain on skin surfaces without penetrating deeper layers. 

NPs have the potential to treat skin diseases such as alopecia, melanoma, and psoriasis [251] Their penetration via intracellular, intercellular, or transcellular pathways are vastly affected by the particle size, composition [250], and skin conditions. Particles with a diameter smaller than 6–7 nm can be absorbed through the lipidic trans-epidermal routes, while those with a size below 36 nm can pass through aqueous pores. Particles of 10–210 nm tend to penetrate through trans-follicular routes. More research is required to fully understand inter-subject variability before they can be suitable for widespread medical use. Particle sizes are affected by encapsulation frequency, zeta potential, molecular weight, and degree of chitosan deacetylation [252]. These parameters can be further studied regarding their effects on the performance of transdermal drug delivery. 

While NPs have shown great promise as drug carriers, it is important to thoroughly evaluate their long-term safety and efficacy. Future research could focus on preclinical and clinical studies to determine the optimal dosages, administration routes, and potential adverse effects of NP-based drugs.

Other research could focus on developing NP-based tools to increase sensitivity and specificity in detecting different types of biomarkers or diseases. Existing advances include NP-based lateral flow immunoassays to detect infectious agents and diseases [253], dextran NPs for diagnosing cardiovascular disorders, fumagillin and docetaxel loading to lipid-based micelles for asthma management [254], NP-based platforms for magnetic resonance imaging (MRI) and photothermal therapy (PTT) tumor imaging [255], and NP-mediated combinatorial phototherapy to chemo-resistant ovarian cancerous tumors [256]. Even so, there is much more to be discovered about NP-assisted diagnostic tools [257].

## Figures and Tables

**Figure 1 biomedicines-11-01124-f001:**
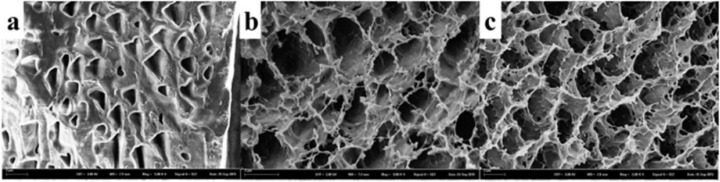
FESEM images of P407/CMCs hydrogel across the skin (**a**) blank hydrogel (PC200), (**b**) 2% P407/CMC loaded hydrogel, (**c**) 4% P407/CMC loaded hydrogel (PC204) (Reprinted from [107] Wang, W. Wat, E. Hui, P.C.L. Chan, B. Ng, F.S.F. Kan, C.W. Wang, X. Hu, H. Wong, E.C.W. Lau, C.B.S. Leung, P.C. Dual-functional transdermal drug delivery system with controllable drug loading based on thermosensitive poloxamer hydrogel for atopic dermatitis treatment. Images reproduced with permission from Scientific Report (2016), https://creativecommons.org/licenses/by/4.0/ (accessed on 16 February 2023)).

**Figure 2 biomedicines-11-01124-f002:**
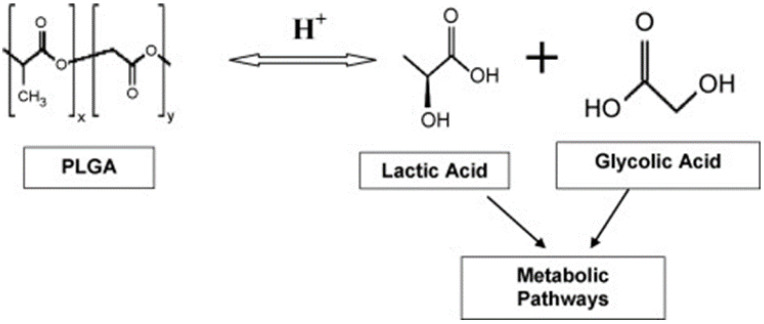
Overview of hydrolysis of poly(lactic-co-glycolic acid) (Reprinted from [117]. Copyright (2010), with permission from Elsevier).

**Figure 3 biomedicines-11-01124-f003:**
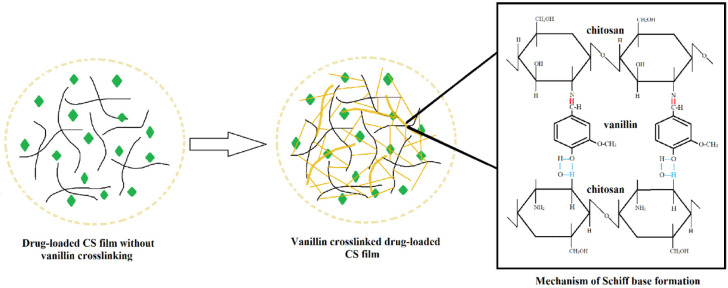
The mechanism of Schiff base formation between chitosan and vanillin (reprinted from [156]. Copyright (2021), with permission of Elsevier).

**Figure 4 biomedicines-11-01124-f004:**
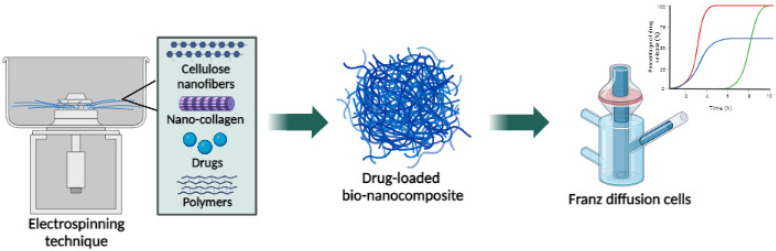
Schematic represents the bio-nanocomposite produced via the electrospinning method and drug release of ketorolac tromethamine via Franz diffusion cells, (Figure generated with Biorender (https://www.biorender.com/, accessed on 13 March 2023).)

**Table 1 biomedicines-11-01124-t001:** The applications of hydrogel in transdermal drug delivery.

Hydrogel	Application	Results	References
Polyethylene glycol diacrylamide (PEG-DA) hybrid hydrogel	Mouse embryonic fibroblast cell lines (NIH 3T3)	Good mechanical property, excellent swelling capacity biocompatibility, and non-toxic to skin	[96]
Gelatin-polyacrylamide (Gel-PAAm) hydrogel	Human skin	Non-toxic to human cells, highly stretchable, and good swelling properties	[89]
Polyacrylamide-grafted-chondroitin sulfate (PAAm-g-CS) hydrogel	Rat abdominal skin	No inflammatory cell infiltration, small degradation of skin, and decreased pore size	[97]
Chitosan-azelaic acid (CS-AZ) hydrogel	L929 mouse fibroblast	Excellent swelling, water vapor permeability, high porosity, and low cytotoxicity	[98]
Carboxymethyl chitosan-silk fibroin peptide/oxidized pullulan (CMCS-SFP/OPL) hydrogel	Newborn porcine skin	Good swelling, water retention properties, skin permeability, water absorption ability, excellent mechanical properties, and biocompatibility	[99]
Carboxymethyl chitosan-grafted-2 hydroxyethyl acrylate (CmCHT-g-pHEA) hydrogel	Micropig dorsal skin	Good pH sensitivity, pores size decreased when ratio of grafting agent increased, improved skin penetration, and non-toxic to skin	[100]

**Table 3 biomedicines-11-01124-t003:** The development of various nanoconjugates from chitosan in the past decade.

Nanocomposite	Transdermal Drug	Findings/Results	References
Polyelectrolyte complexes (PEC) with carboxymethylagarose (CMA) and chitosan (CS) as pH-responsive carriers	Diclofenac sodium (DS)	Immortalized human keratinocyte (HaCat) cells showed approximately 100% survival with 67% cumulative drug release after 72 h at 37 °C and pH 6.0 through the Fickian diffusion mechanism.	[159]
Chitosan microneedle patches (85% deacetylated, molecular weight: 1526.464 g/mol)	Meloxicam	A higher concentration of acetic acid displayed greater resistance to compressive force as temperature increased and the penetration study indicated sustained insertion of microneedles in cow’s ear cadaver skin.	[160]
Chitosan/hyaluronan transdermal film	Thiocolchicoside	Easy and reliable administration with high efficiency in drug release; flexible dosage, minimal drug dosage/frequency to reduce side effects.	[161]
Chitosan and phytagel (gellan gum) transdermal hydrogel	Ibuprofen	Chitosan improved the drug permeability to skin and increased the transdermal release rate of ibuprofen by a factor of 4.	[162]
Chitosan/phospholipids nanofibers	Curcumin, diclofenac, and vitamin B12	Cytotoxicity studies confirmed the good biocompatibility of the nanofibers, the drug release rate relied eminently on the drug solubility.	[163]
Carboxymethyl chitosan-grafted-2 hydroxyethyl acrylate (CmCHT-g-pHEA) hydrogel	Micropig dorsal skin	Good pH sensitivity, pores size decreased when ratio of grafting agent increased, improved skin penetration, and non-toxic to skin	[100]
Carboxymethyl chitosan/oxidized pullulan hydrogel-based microneedles	Salvia miltiorrhiza extract	Good mechanical strength, high water absorbing capacity, good skin permeability, and rapid drug release into the targeted porcine skin.	[99]
N-methacryloyl chitosan (N-MAC) microgels	Bovine serum albumin (BSA)	High cell viability in N- MAC hydrogel. Rapid transdermal curing hydrogels (in vivo) for localized and sustained protein delivery.	[164]
N,N,N-trimethyl chitosan (TMC), polyethylene glycolate hyaluronic acid (PEG-HA), and polysaccharide-based nano-conjugate of hyaluronic acid, chitosan oligosaccharide and alanine [HA-Ala-Chito(oligo)]	Chinese medicine CortexMoutan (CM)	The ex vivo transdermal release results showed significant drug permeability into the skin. The MTT assay results showed high cell viability of human HaCaT keratinocytes, suggesting no cytotoxicity on skin cells.	[165]
Chitosan-coated poly(dl-lactide-co-glycolide) (PLGA) nanoparticles	Donepezil hydrochloride (DP)	Chitosan-coated PLGA nanoparticles delivered drugs to the deep hair follicles more efficiently through iontophoretic transdermal delivery, as compared to the bare PLGA nanoparticles.	[166]
Polyvinyl alcohol-Chitosan (PVA/CS) bioconjugate	Colchicine	Significant colchicine deposition in the skin with remarkable cytotoxicity against a melanoma cell line.	[167]
Chitosan-coated nanoemulsion (NE2-CS), uncoated nanoemulsion (NE1), and quaternized chitosan (QCS)	Zingiber cassumunar Roxb (Plai extract)	QCS improved the stability and transdermal properties of the Plai extract, as compared to NE1 and NE2-CS. NE2-QCS showed higher cytotoxicity to the breast (BT474) and oral cavity (KB) cancer cell lines than the Plai extract alone and had 1.5-fold higher permeability and cumulative release of the Plai extract than NE1.	[168]
Chitosan sponges	Hormonal drug 17*β*-estradiol (E2) with a purity of 99%	High drug loading was reported. Uniform distribution of E2 crystallites in the chitosan sponge volume was observed, improving the bioavailability of the drug.	[169]
PLGA chitosan transdermal Pluronic nanogel	Temozolomide	The in vitro drug release showed 85% transdermal release at a mildly acidic pH mimicking the skin microenvironment. Ex vivo studies displayed a penetration rate with 80% Temozolomide uptake in porcine epidermal tissue.	[170]
Carboxymethyl chitosan/2-hydroxyethyl acrylate hydrogel	Nobiletin	Mechanism of the nobiletin from the hydrogel was confirmed to be Fickian diffusion. In vitro skin permeation experiments showed that the hydrogel improved the transdermal delivery of nobiletin.	[100]
ZnO nanorods with chitosan hydrogels crosslinked with azelaic acid	Acetylsalicylic acid	The controlled drug release behaviors of nanocomposites according to the first-order kinetic model and was confirmed to be non-toxic to L929 mouse fibroblasts by XTT assay.	[98]
Chitosan nanoparticles mucoadhesive gel	Propranolol hydrochloride	High encapsulation efficiency and drug loading improved systemic bioavailability and therapeutic efficacy of propranolol-HCl in a transdermal delivery system. Thixotropic behavior with prolonged drug release properties was observed.	[171]

## Data Availability

Not applicable.

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
