# Peer review of "Recent Development of Nanomaterials for Transdermal Drug Delivery"

_biomedicines, 2023, doi:10.3390/biomedicines11041124_

Round 1

Reviewer 1 Report

The authors have written a comprehensive review article on “Recent Development of Nanomaterials for Transdermal Drug Delivery” (biomedicines-2268328) which is impressive and add value to the field of novel drug delivery systems. I would recommend the acceptance of the publication of this manuscript after a minor revision which is as follows:

-        Authors are suggested to add a section describing the future prospects.

Author Response

Thank you for your kind suggestion. The future prospects were added in page 16.

Reviewer 2 Report

The review article about transdermal drug delivery systems based on nanomaterials has explained the recent research activities carried out related to that. The article introduced all the possible materials for transdermal drug delivery systems and their advantages over other drug carriers. From the studies, it was concluded that at present, the most promising NP-based carriers are polylactic acid (PLA-NP) or polyethylene glycol (PEG-NP). The studies also reveal the possibility of producing different composites. The nanomaterial-based transdermal drug delivery systems are found to be biocompatible, non-toxic, highly stable, biodegradable, and have high loading capacities, good storage stabilities, higher drug encapsulation efficiency, and sustained and controlled drug release.

The structure of the manuscript is good. 

Suggestions

1.     Line 118, rewrite the chemical symbol HCuSNPs.

2.     Addition of a preposition from, for indicating the range of average intracellular delivery.

3.     Rewrite sentences 224 and 227

4.     Line 480, put ILs in parenthesis.

5.     Line 581, the spelling of matrics.

6.     The article can add graphical representation data of drug release.

Author Response

Thank you for your kind suggestions. Please find our point-by-point response below:

1. The HCuSNPs is rewritten in Line 118

2. The preposition from is added in Line 206

3. The sentence in Line 224 is rewritten as ‘’The skin permeability of rat abdomen skin was studied by using the antisolvent diffusion method to create indomethacin (IM)-loaded PGLA NPs’’ and Line 227 ‘’ The use of iontophoresis resulted in a greater penetration of 50 nm PLGA nanoparticles into rat skin as compared to 100 nm PLGA nanoparticles’’

4. The (ILs) is replaced in Line 487

5. The spelling of matrices is corrected in Line 587

6. The graphical drug release is added in Line 444 (Figure 4)

Extra

1. An image is added in Line 284 with a description (Figure 3). The mechanism of Schiff base formation between chitosan and vanillin